# Numerical Analysis of the Seismic Performance of Light-Frame Timber Buildings Using a Detailed Model

**Franco Benedetti** [1,2,*] [ID]**, Alan Jara-Cisterna** [1]**, Juan Carlos Grandón** [3]**, Nicool Astroza** [1] **and Alexander Opazo-Vega** [1,2] [ID]

1 Department of Civil and Environmental Engineering, University of Bío-Bío, Concepción 4081112, Chile; afjara@ubiobio.cl (A.J.-C.); astrozanicool@gmail.com (N.A.); aopazove@ubiobio.cl (A.O.-V.)
2 Centro Nacional de Excelencia para la Industria de la Madera (CENAMAD), Pontifical Catholic University of Chile, Santiago 7820436, Chile
3 Department of Structural and Geotechnical Engineering, Pontifical Catholic University of Chile, Santiago 7820436, Chile; jcgrandon@uc.cl
* Correspondence: fbenedet@ubiobio.cl; Tel.: +56-41-3111646

**Abstract:** Timber structures have gained interest for the construction of mid-rise buildings, but their seismic performance is still a matter under development. In this study, a numerical analysis of the seismic performance of light-frame timber buildings is developed through a highly detailed model using parallel computing tools. All of the lateral-load-resisting system components and connections are modeled. Combinations of lateral load capacity distributions in structures of one, three, and five stories are studied in order to assess the effects on the global performance of different triggered failure modes through nonlinear static and dynamic analyses. The results suggest that shear bracket connections and sheathing-to-framing connections control the buildings' responses, as well as the failure mode. For a ductile response, the lateral displacement must be dominated by the in-plane wall distortion (racking); therefore, the system must be provided with a story shear sliding stiffness and load capacity at least twice that of the walls. Furthermore, based on the pushover capacity curves, the performance limits are proposed by evaluating the stiffness degradation. Finally, the effect of the mobilized failure mode on the structural fragility is analyzed. Even though standard desktop PCs are used in this research, significant reductions in the computation effort are achieved.

**Keywords:** light-frame timber buildings; seismic performance; detailed nonlinear finite element modeling



## 1. Introduction

Mid-rise wooden light-frame buildings have been widely used in high-seismic-risk regions. The high deformation capacity inherent in wood elements and joint connections (framing-to-framing, sheathing-to-framing, etc.) has demonstrated good structural performance under large seismic demands [1–3].

As with other light-weight structural systems, light-frame timber buildings take advantage of the low seismic lateral loads that are induced [4–6]. In timber light-frame buildings, the typical lateral-force-resisting system is formed by the timber shear walls. The lateral load behavior of the wooden frame walls is a complex phenomenon that involves the interaction of several components that produce different deformation and failure mode mechanisms. The study of this complex behavior using simplified numerical models has been widely accepted due to the low computation effort required and the demonstrated capabilities to reproduce specific experimentally observed responses. One of the first attempts to reproduce the hysteretic behavior of wood shear walls through numerical models was developed by Dolan [7] and Kasal and Xu [8]. They concluded that the sheathing-to-framing connections govern the timber shear walls' load capacity, stiffness, and ductility.

Similarly to other structural systems as domes [4,5] and concrete frames [6], a complete evaluation of the seismic response of wooden structures requires the development of nonlinear models able to reproduce the inelastic behavior of the materials and connections, as well as the contact nonlinearities and hysteretic energy dissipation. Folz and Filiatrault [9] presented a simplified numerical model to represent the hysteretic response of a single timber shear, wall which can also be used in the analysis of multi-story buildings. This numerical model formulation was incorporated into the CASHEW software [10] through the SAWS constitutive model [9]. Moreover, Folz and Filiatrault [11] evaluated the accuracy of the SAWS model, presenting a comparison of the numerical model and a full-scale test of a two-story wood-framed house. The numerical model showed a good capability to reproduce the relative displacement obtained through experimental tests. Nevertheless, the numerical model was not able to properly capture the torsional response, and its calibration is only based on the sheathing-to-framing connection load–displacement response. The latter simplification suggests that the model is based only on the racking mechanism of the wall, while other failure modes are neglected or disregarded.

Pei and van de Lindt [12] developed a shear–bending formulation for wooden light-frame systems that is able to reproduce the global experimental response of light-frame wall systems. Their findings suggest that the cumulative uplift and the out-of-plane rotation of the horizontal diaphragm need to be considered for three-story buildings and higher. Furthermore, the authors noted that the behavior of the stacked shear walls in a building differs from the response of an isolated wall.

Pei and van de Lindt [13] presented a comparison of a numerical simulation and the experimental seismic response of a six-story wooden light-frame building using the coupled shear-bending formulation [12]. The numerical simulation showed a slight underestimation in the inter-story displacement. Moreover, the authors mentioned that the numerical model does not adequately capture the torsional response.

Subsequently, Pang et al. [14] proposed the timber 3D model, which is a three-dimensional extension of the bi-dimensional shear wall models developed by Christovasilis and Filialtraut [15] and Pang and Shirazi [16]. In this formulation, the floor diaphragm is defined by 12 DOF that take into account the in-plane and out-of-plane flexibility. This model can capture the timber light-frame buildings' collapse mechanism and has been used to perform incremental dynamic analyses to a three-story building. Furthermore, Pan et al. [17] used a 2D model (corotational beam element with 6 DOF included in the M-CASHEW2 software [18]) and a 3D model (timber 3D) to reproduce the observed behavior of a shake table test of a two-story light-frame building. The results showed good accuracy in reproducing the global response on both numerical models.

All of the aforementioned simplified models have been accepted and validated by the academic and professional communities. They have been widely used for the numerical study of the seismic performance of light-frame timber buildings because this strategy is numerically efficient; thus, the computation effort remains manageable. In contrast, the major drawback of the simplified models is that the intrinsically complex mechanism of the lateral load response of the structural system may be oversimplified; therefore, the results and findings can include uncertainties and limitations, since the effects of the three-dimensional coupling, shear displacements, and vertical loads are not properly assessed. On the other hand, the computational and simulation capabilities available nowadays bring about opportunities to improve the analyses by implementing more complex models that are able to reproduce the complete lateral load response mechanism of light-frame structures.

Recent efforts have been made to develop nonlinear models that incorporate a larger number of deformation mechanisms or specific conditions. Di Gangi et al. [19] proposed a more detailed modeling approach applied to single light-frame timber shear walls. They modeled each sheathing-to-framing connection, the timber frame elements, and the sheathing board. This model was used to perform a parametric study to characterize variables that influence the racking capacity of a timber shear wall. In addition, Kuai et al. [20]

developed a detailed finite element modeling strategy to analyze the different deformation behaviors that a light-frame wall can generate. They concluded that the detailed approach can reproduce the experimentally observed local response with reasonable accuracy. Other specific conditions that have been assessed are the detailed modeling of high-strength wood-framed shear walls [21], and the development of simplified models that include the vertical load and bending moment effects [22]. However, these efforts were implemented at the wall element level, and not all connections or components were included in the modeling. Their application to the analysis of buildings is still under development [20].

The study of the lateral load behavior of other structural systems that exhibit a similar seismic response to light-frame timber buildings (e.g., cross-laminated timber buildings) has also highlighted the need for a local and detailed approach to the modeling and analysis in order to identify all of the response mechanisms [23–27]. Consequently, through detailed models, Shahnewaz et al. [27] developed a seismic fragility analysis of cross-laminated timber.

Given the described limitations of the current modeling methods applied to light-frame timber buildings, it is necessary to develop advanced strategies in order to achieve a better comprehension of the seismic behavior of these structures. Consequently, this work explores the seismic response of mid-rise light-frame timber buildings through complex and detailed numerical models implemented using parallel computing in an open-source finite element tool. The implemented novel modeling approach allows the incorporation of all lateral-load-resisting system components simultaneously with the effects of the vertical load and three-dimensional coupling. Hence, the complex nature of the different nonlinear deformation mechanisms that light-frame buildings exhibit is explicitly considered in the analysis.

The developed seismic response assessment involves a parametric study in which the different lateral-load-carrying conditions are varied to evaluate their effects on the seismic behavior. The research involves the evaluation of the lateral load behavior through nonlinear static and dynamic analyses. As a consequence, the results of this study are expected to contribute to a deep understanding of several nonlinear structural phenomena in the response of light-frame timber buildings, such as the effect of the lateral load capacity distribution on the global failure mode, the seismic performance limit states, and the fragility under large and long-duration seismic demands.

## 2. Materials and Methods

### 2.1. Studied Building Configuration

This research involved the study of timber housing buildings with one, three, or five stories, depending on the analysis performed. The plan dimensions of the building were 24.1 m in length by 12 m in width, with 2.44 m of inter-story height. The floor plan of the archetype is shown in Figure 1, where the red arrow represents the orientation of the load transfer direction of the one-way floor system. The total heights of the building floors were 2.44 m, 7.32 m, and 12.2 m for one, three, and five stories, respectively. The research archetype involved 3.1% and 5.1% wall densities in the longitudinal and transverse directions related to total wall lengths of 59.1 m and 97.1 m, respectively.

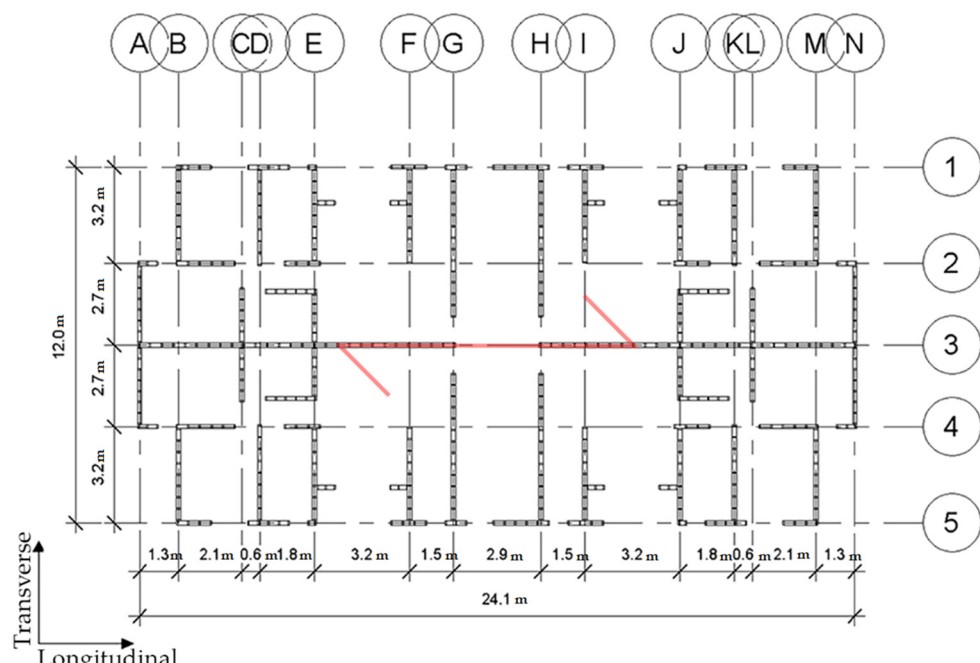

**Figure 1.** Plan view of the studied archetype. The red arrow represents the orientation of the load transfer direction of the one-way floor system.

For the structural design, the Chilean seismic code NCh 433:1996 [28], the Chilean structural wood design standard NCh 1198:2014 [29], and the ANSI/AWC SDPWS-2015 code were used, considering an allowable stress design approach (ASD) [30]. The design was developed using standard force-based methods that do not promote any particular failure mode. However, for the sake of the purposes of this research, the design aimed to prevent the rocking response by controlling the system's failure mechanism. Regarding the seismic design, according to NCh 433:1996, the lateral load demand is defined considering a lateral load pattern using the maximum seismic coefficient $C = 0.208$ recommended for timber buildings with a load reduction factor $R = 5.5$. The seismic weight is determined as the dead load plus 25% of the live load, resulting in a 2 kN/m$^2$/story, which considers lightweight concrete slabs over the timber floors.

The typical shear wall configuration we used is presented in Figure 2. Considering the high levels of tension and compression forces on the structural elements transmitted by the upper floors, multiple studs were employed at the edges of the wall. MGP10-graded Chilean radiata pine [29], with a cross-section of 45 mm × 142 mm, was used for the studs and plates. For the sheathing boards, oriented strand boards of 11.1 mm thickness nailed to the timber frame with 2.94-mm-diameter helical nails were used. The nailing pattern considers nails spaced at 50 mm on the edges of the board and 100 mm for the inner studs.

Regarding the wall anchors, hold-down devices and shear brackets were added following the structural design requirements. At the foundation level, the connectors were considered to be anchored to concrete. Moreover, a steel rod through the floor elements was used for the upper levels to connect the hold-downs between adjacent stories.

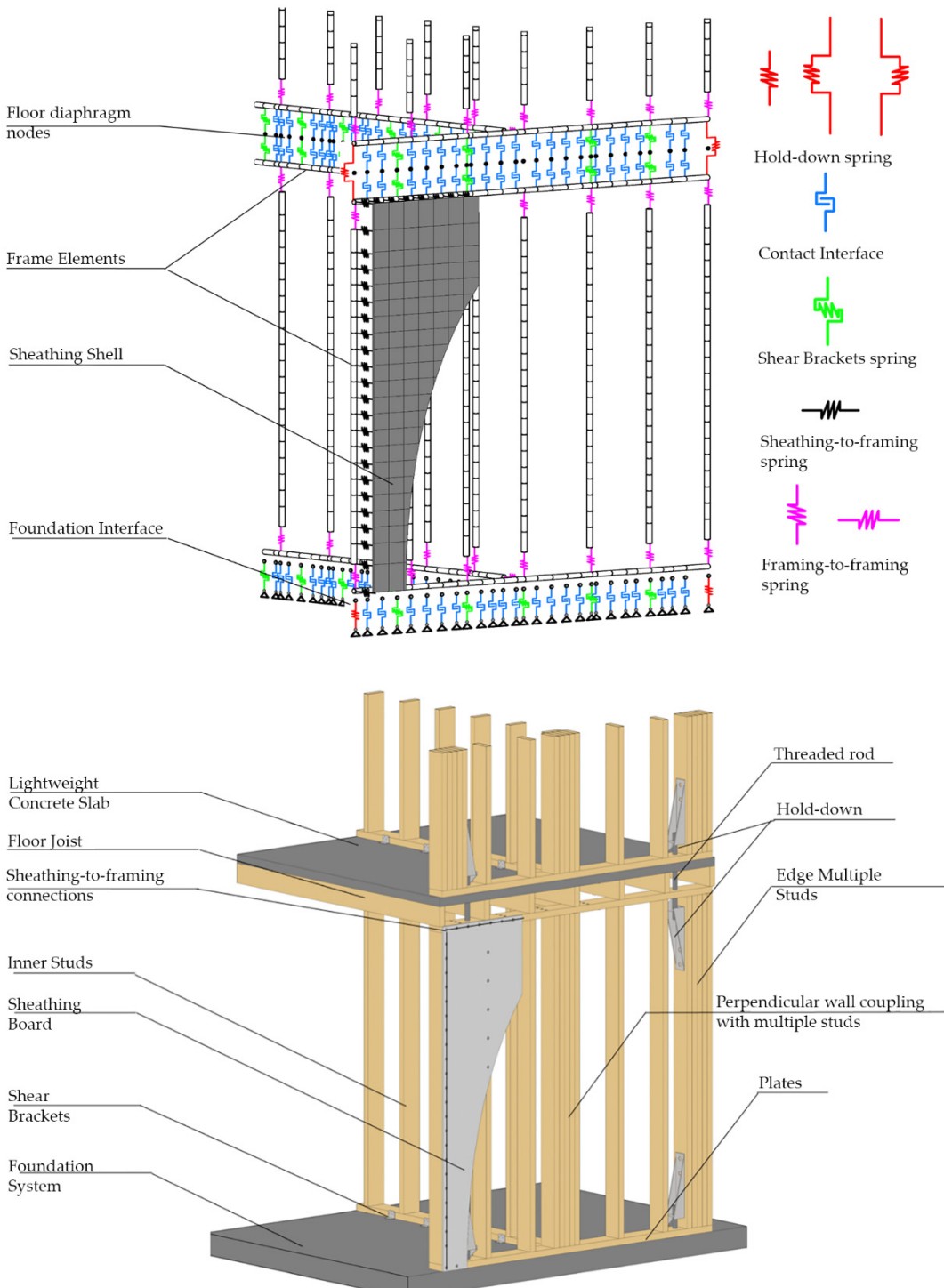

**Figure 2.** Light-frame timber shear wall components (**below**) and model implementation (**above**).

### 2.2. Modeling Approach

This study involves the development of a set of three finite element models for the building described previously based on the design case conditions. Here, a model called M1 is used to analyze the behavior of the building measuring one story, the M3 model is used to study the response considering three stories, and the M5 corresponds to the building with five stories. For each of these models, different conditions are studied through the variation in the properties of the components at the local level. Nevertheless, the modeling approach is the same for all models.

Detailed coupled three-dimensional numerical models were developed in OpenSees [31]. Every component of the timber light-frame wall is included (studs, sheathing boards, plates, nails, and mechanical connections), as well as the friction and contact interactions at the foundations and floor slabs. Figure 2 shows the wall components and the corresponding finite element model.

According to the expected failure mechanisms, the models consider the linear and nonlinear components. Elastic beam–column elements are used for the timber frame members (studs and plates), elastic isotropic shell elements for the OSB sheathing boards, and linear elastic springs for the timber-to-timber joints (stud-to-plate connections). Moreover, the nonlinear behavior is assigned to the components of the system where the damage and energy dissipation are likely to occur. The sheathing-to-framing, shear, and uplift connections are modeled using hysteretic nonlinear springs. At the same time, for the kinematic interactions between walls and foundations and between walls and floors, a contact interface is employed. Table 1 summarizes the general modeling properties.

**Table 1.** Modeling properties of elements and connections.

| Type | Element | Characteristics | Property | OpenSees Element–Constitutive Material |
|---|---|---|---|---|
| Linear | Studs and plates | 45 mm × 142 MGP10 radiata pine lumber | E = 10,200 MPa [29] | elasticBeamColumn–Elastic |
| | Sheathing boards | 11.1 mm thickness OSB | E = 3485 MPa [32] G = 1383 MPa [32] | ShellMITC4–Elastic Isotropic |
| | Timber-to-timber connections | Annular shank nails | Table 2 [33] | zeroLength–Elastic |
| Nonlinear | Sheathing-to-framing connections | Helical nails | Table 2 [34] | twoNodeLink–Pinching4 |
| | Shear connections | Angle-bracket device | Table 2 [35] | twoNodeLink–Pinching4 |
| | Hold-down connections | Hold-down anchor | Table 2 [35] | zeroLength–Pinching4 |
| | Contact interface | Timber-to-timber friction | Friction coefficient = 0.5 [36] | zeroLengthContact3D |

**Table 2.** General mechanical properties of connections in the models.

| Connection | Yield Displacement $\Delta_y$ (mm) | Ultimate Displacement $\Delta_u$ (mm) | Ductility Capacity $\mu$ | Load Capacity $F_{max}$ (kN) | Elastic Stiffness $K_e$ (kN/mm) |
|---|---|---|---|---|---|
| Hold-downs (tension) | 13.0 | 24.0 | 1.84 | 106.3 | 7.7 |
| Shear brackets (shear and tension) | 10.0 | 24.8 | 2.48 | 21.6 | 2.0 |
| Sheathing-to-framing (parallel and perpendicular to grain) | 1.3 | 12.0 | 9.23 | 1.7 | 0.95 |
| Timber-to-timber | - | - | - | - | 0.25 |

With respect to the mesh refinement, the size of the structural component elements (shells and frames) is defined by the spacing of the joints and connections, particularly by the sheathing-to-framing connections spaced at 10 cm at the border of each OSB board.

For the timber-to-timber connections, just the withdrawal and shear stiffness are included in the model due to the scarce contribution of the rotation stiffness component to the global wall lateral response [37]. Regarding the hold-downs, the model considers only the tension stiffness, while the shear component is discarded. Moreover, to approximate the coupled response of the two principal directions of the sheathing-to-framing and shear bracket connections, the model is idealized with two orthogonal nonlinear springs to take into account the parallel- and perpendicular-to-grain responses of the sheathing-to-frame nails, as well as the shear and tension stiffness for the shear brackets. For the shear brackets case, the shear and tension behavior are assigned to be equal, supposing that the

fasteners of the connection control the response. Concerning the contact interface, it is modeled to permit the sliding and uplifting movement of the wall and to reproduce the frictional interactions with the foundation and floor slabs. The contact modeling includes an auxiliary layer of dummy nodes to solve numerical issues related to the compatibility of the DOFs of the contact movement and the structural components. For the base nodes, at the foundation level, a fixed constraint is applied for the three translation degrees of freedom. The mechanical properties of the connections are summarized in Table 2.

It is important to mention that the connection parameters are defined from different literature studies, except for the sheathing-to-framing connection, the data for which are obtained through experimental tests. Using the experimental and literature data, the pinching4 material law [38] is calibrated for the modeling of each connection nonlinear spring. The pinching4 parameters of every mechanical connection and the experimental test information can be found in Appendices A and B.

Regarding the floor elements, their modeling strategy tends to reduce the computational effort. Just the nodes, masses, and gravitational loads are considered in the model, while the floor's structural members (e.g., joists, floor boards, and lightweight concrete slabs) are not physically included. Finally, considering the small aspect ratio of the floor plan and the additional in-plane stiffness provided by the lightweight concrete slab, a rigid diaphragm constraint is assigned to the floor nodes of every story.

As this research comprises the execution of nonlinear analyses for the evaluation of the seismic behavior, a sequential loading procedure is used. The static pushover and time history analyses are developed after the solution of the system under vertical loads. This staged procedure is crucial for developing the initial state of the structure, particularly for the contact interface elements. Additionally, since the elastic damping is complex to accurately include in parallel nonlinear time history analyses, the only source of damping considered is the hysteretic energy dissipation provided by the yielding of nonlinear components and the friction.

### 2.3. Model Implementation and Parallel Segmentation

Due to the large size of the detailed models and their high computation demand (e.g., about 1,900,000 DOF for model M5), parallel computing techniques are used. This computational tool allows the model to be segmented into different domains to take advantage of the processing cores available on the CPU. Therefore, the parallel multi-process version of OpenSees is employed (OpenSeesMP).

As the simulation tool does not have a graphical interface, the models are preprocessed in SAP2000 software, where the geometry, masses, and forces are defined for later conversion into the OpenSees language. Figure 3 shows a view of the M1, M3, and M5 geometrical models developed in SAP2000.

The analysis using parallel computing techniques requires the segmentation of the physical model into a finite quantity of subdomains to distribute the computational load among the CPU cores. Several strategies to segment nonlinear computational domains have been developed (e.g., [39,40]), but in this study a static decomposition approach is used because it is easy and straightforward. However, the static decomposition approach can be inefficient for highly nonlinear problems, and dynamically adaptive techniques are preferable.

MATLAB routines were developed to subdivide the physical and geometrical model into substructures, replicating common parts but not elements that can generate the superposition of effects on the response of the analyzed system (e.g., forces, masses). In this work, four desktop PCs with eight-core Intel i7 processors were used to perform the analyses; therefore, the parallelization considers the segmentation into eight subdomains.

To analyze parallel models, matrix analysis algorithms that simultaneously solve the systems of equations in all the processor cores must be used. Thus, the powerful parallel solver MUMPS (multifrontal massively parallel sparse direct solver) is employed.

Moreover, the parallel reverse Cuthill-McKee (Parallel RCM) scheme is selected to number the DOFs and order the matrix equations.

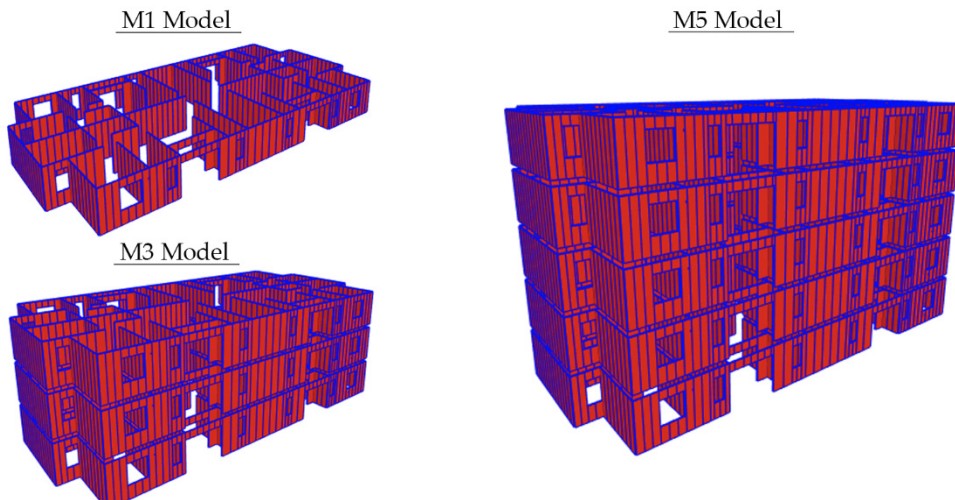

**Figure 3.** View of the M1, M3, and M5 models developed in SAP2000.

Moreover, since the contact elements used in the wall–foundation and wall–floor interface modeling tend to generate spurious high-frequency content during time history analyses [41], the TRBDF2 [42] implicit time integration scheme is used because it filters the generated noise by conserving energy and momentum at every time step.

Since the convergence and numerical stability are not guaranteed in nonlinear modeling, a simple approach is implemented to handle the convergence troubles. The nonlinear solving algorithm, time step (or displacement step), and convergence tolerance will be dynamically and adaptively modified if necessary.

*2.4. Seismic Response Analysis Scenarios*

Different analysis scenarios are next developed based on the design structure by means of modifications of the load-carrying capacity of the local components of the system. Using the models M1, M3, and M5, the effects of the stiffness and load capacity distribution among the lateral-load-resisting system in the global failure mode and the seismic response is studied through nonlinear static and dynamic analyses.

2.4.1. Nonlinear Pushover Analysis

Through displacement-based static pushover analyses and a parametric evaluation, models M1, M3, and M5 are employed for several purposes.

- Single-story failure mode assessment.

Model M1 is employed to define how the lateral displacement in a single-story structure is provided between two different deformation mechanisms: wall distortion and base sliding. Even though the developed models are capable of reproducing the wall rocking behavior, in the analysis performed in this research, its effect is not assessed. Consequently, the conditions that control the rocking motion (e.g., hold-down connection properties) are not varied among the studied cases.

For the single-story system, Figure 4 presents the three global failure modes considered. The first is mode 1, which considers that the total lateral roof displacement $\Delta_T$ under a horizontal load $F_H$ is mainly provided by the base sliding mechanism ($\Delta_S$), while the wall distortion displacement ($\Delta_W$) is negligible. The opposite behavior is defined for mode 3, where $\Delta_T$ is now attained principally by $\Delta_W$, while $\Delta_S$ is very small (promoting a racking-dominant behavior). Furthermore, a combined mode (mode 2) is defined when both $\Delta_W$ and $\Delta_S$ are large enough, and neither of them can be discarded.

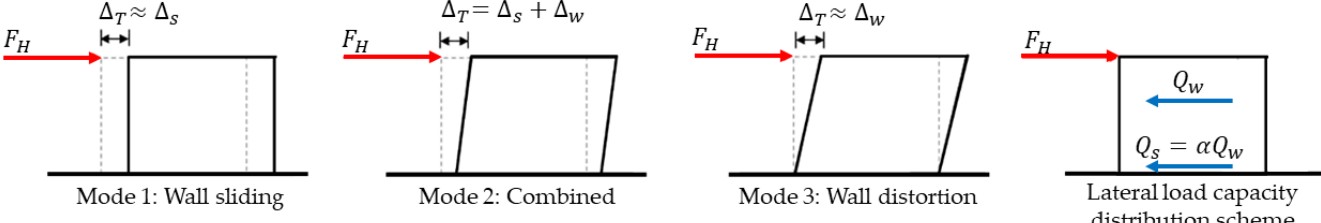

**Figure 4.** Single-story (M1 model) global failure modes and lateral load capacity distribution scheme.

Regarding the relation between the two lateral deformation mechanisms, the factor $\alpha = \frac{Q_S}{Q_W}$ is defined, where $Q_S$ represents the story sliding load capacity provided by the shear brackets, while $Q_W$ stands for the story in-plane load wall capacity supplied by the sheathing-to-framing connections. Figure 4 provides a schematic representation of the lateral load capacity distribution. Five study cases are defined for the M1 model using $\alpha = [0.5, 1.0, 2.0, 3.0, 4.0]$. These factor values are achieved by modifying the story sliding capacity through variations of the shear brackets design while the story in-plane wall capacity is kept constant.

- Inter-story deformation distribution.

The M3 model is employed to examine the effect of the lateral load capacity distribution in height on the story displacements. For the capacity curve calculations, the lateral load patterns considered are a lateral load pattern according to the seismic force demands of the Chilean seismic code [28] and of the ASCE 7–16 code [43], an inverted triangular distribution, and a distribution that promotes a first vibration mode equivalent deformed shape. Two lateral load capacity conditions are studied. The first one assumes that the load capacity is uniform in all stories, being defined by the first story's design configuration (this condition is defined as $\beta$ = uniform). The second condition supposes that every story is designed with a load capacity proportional to the respective lateral force demand, so the sheathing-to-framing connection load response is adjusted to match the imposed demand (stated as $\beta$ = variable condition). The $\beta$ = uniform condition is studied through case M3-1, while the $\beta$ = variable is analyzed in case M3-2. For both conditions, an $\alpha = 2$ factor is considered for all stories; hence, the story sliding capacity $Q_{si}$ is twice the story in-plane wall strength $Q_{wi}$. Figure 5 shows the schematic representations of the lateral load capacity diagrams for the studied conditions.

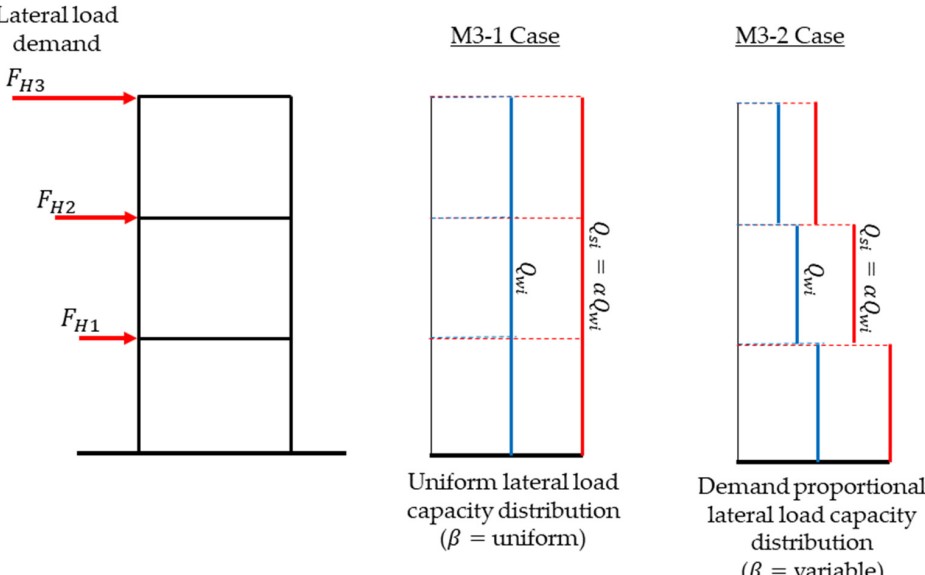

**Figure 5.** Lateral load capacity distribution scenarios for the M3 model.

- Global damage limit states.

The effect of the load capacity distribution in the global limit states is assessed through static pushover analyses of the M5 model. Four cases are defined, where the findings for the failure mode control (obtained with M1 and M3 analyses) are evaluated by analyzing different combinations of the $\alpha$ factor and the lateral load capacity distribution in the structure's height conditions ($\beta$). Two values are considered for the $\alpha$ factor to promote a combined failure mode or a wall-distortion-controlled mode. These $\alpha$ ratios are defined as equal for all stories. Regarding the $\beta$ condition, it is considered uniform and variable to trigger either a soft-story mode or a balanced inter-story drift demand. Finally, pushover analyses are performed under a lateral load pattern defined by the design code's seismic force demand [28]. In Table 3, the details of all studied cases are shown.

**Table 3.** Studied case conditions.

| Case | $\alpha = \frac{Q_s}{Q_w}$ | Load Capacity Distribution in Height $\beta$ |
|------|------------|----------------------------------------------|
| M1 | 0.5, 1.0, 2.0, 3.0, 4.0 | - |
| M3-1 | 2.0 | Uniform |
| M3-2 | 2.0 | Variable |
| M3-D | 2.0 | Variable |
| M3-B | 0.5 | Variable |
| M5-1 | 2.0 | Variable |
| M5-2 | 0.5 | Variable |
| M5-3 | 2.0 | Uniform |
| M5-4 | 0.5 | Uniform |

### 2.4.2. Incremental Dynamic Analysis

The incremental dynamic analysis [44] technique (IDA) is implemented to develop a seismic fragility analysis. Aiming to keep the computational cost manageable, the M3 model is employed. To assess the effect of the failure mode in the seismic safety, two different cases are defined using the modeling capabilities. First, a low-deformation-capacity model is defined through the combination of $\alpha$ and $\beta$ conditions. This case is called the brittle model (M3-B). For the second model, named the ductile model (M3-D), the $\alpha$ and $\beta$ factors are selected to promote a large deformation capacity response. The defined cases are presented in Table 3.

Large Chilean earthquake recordings are used for the IDA seismic demands. The seismic response of the structural models is evaluated under six acceleration time series of the horizontal components of the $M_w$ 7.1 Punitaqui 1997, $M_w$ 7.7 Tocopilla 2007, and $M_w$ 8.8 Maule 2010 earthquakes recorded at Illapel, Tocopilla, and San Pedro stations, respectively. Figure 6 shows the acceleration time series and pseudo-acceleration spectra of the employed seismic demands. For the IDA application, every motion is normalized and scaled according to FEMA P695 [45] recommendations.

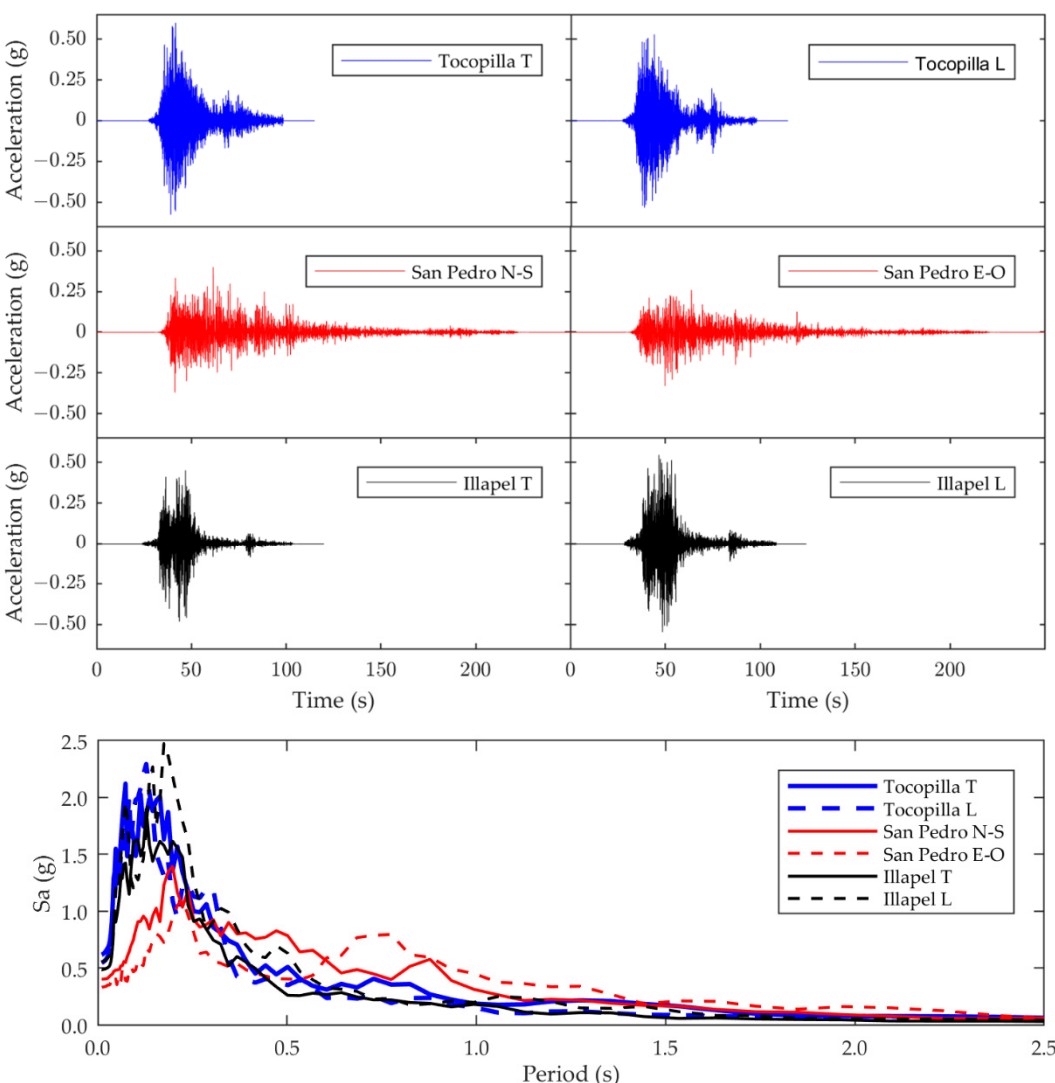

**Figure 6.** Seismic demands used in the incremental dynamic analyses.

As observed in Figure 6, the ground motions reach PGAs of 0.6 g, 0.4 g, and 0.56 g for the Tocopilla, San Pedro, and Illapel records, respectively. Moreover, the ground motions' significant durations are 33.59 s, 72.81 s, and 18.81 s for Tocopilla, San Pedro, and Illapel, respectively. In terms of the pseudo-accelerations, it can be observed that the Tocopilla and Illapel spectra achieve higher pseudo-accelerations for periods between 0.06 s and 0.23 s, while in the case of San Pedro recording, the spectrum achieves higher pseudo-accelerations for periods within 0.16 s to 0.87 s.

### 3. Results and Discussion

This chapter discusses the seismic analysis results for the light-frame timber buildings obtained through detailed models. These results provide an initial analysis of the capabilities of the highly detailed models for timber structures. A total of 13 models are developed to study the different cases defined here, and their conditions and configurations are presented in Table 4.

**Table 4.** In-plane wall capacity per story and analysis direction.

| Case | In-Plane Load Wall Capacity $Q_w$ (kN) | | | | | | | | | |
|---|---|---|---|---|---|---|---|---|---|---|
| | 1st Story | | 2nd Story | | 3rd Story | | 4th Story | | 5th Story | |
| | Long. | Trans. | Long. | Trans. | Long. | Trans. | Long. | Trans. | Long. | Trans. |
| M1 | 582 | 865 | - | - | - | - | - | - | - | - |
| M3-1 | 582 | 865 | 582 | 865 | 582 | 865 | - | - | - | - |
| M3-2 | 582 | 865 | 475 | 705 | 336 | 500 | - | - | - | - |
| M3-D | 582 | 865 | 475 | 705 | 336 | 500 | - | - | - | - |
| M3-B | 582 | 865 | 475 | 705 | 336 | 500 | - | - | - | - |
| M5-1 | 970 | 1441 | 850 | 1260 | 730 | 1080 | 575 | 855 | 425 | 630 |
| M5-2 | 970 | 1441 | 850 | 1260 | 730 | 1080 | 575 | 855 | 425 | 630 |
| M5-3 | 970 | 1441 | 970 | 1441 | 970 | 1441 | 970 | 1441 | 970 | 1441 |
| M5-4 | 970 | 1441 | 970 | 1441 | 970 | 1441 | 970 | 1441 | 970 | 1441 |

### 3.1. Global Failure Mode Control

The effect of the lateral deformation mechanism on the global failure mode is shown in Figure 7, where the M1 model pushover curves in the transverse direction are presented in terms of the ratio between the lateral load applied ($F_H$) and the weight of the structure ($W$) against the story drift for each $\alpha$ factor considered. In every plot, the total lateral drift curve is displayed along with the distribution of this deformation into the two displacement mechanisms defined.

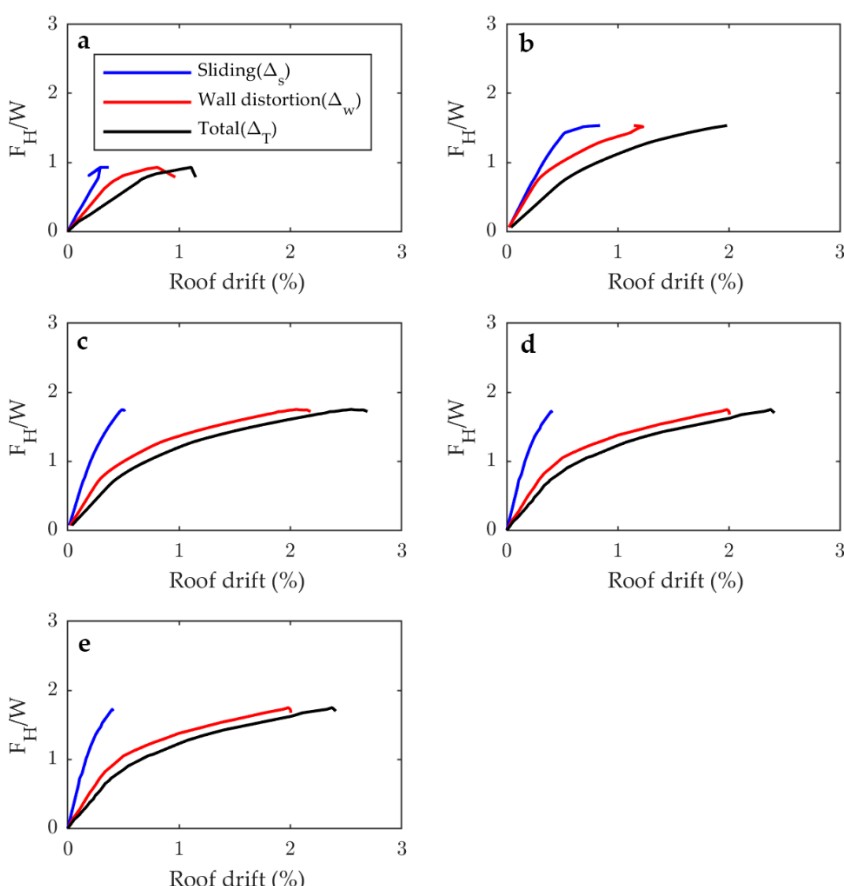

**Figure 7.** Effects of the alpha index on the contributions of wall sliding and distortion in the story deformation: (**a**) $\alpha = 0.5$, (**b**) $\alpha = 1.0$, (**c**) $\alpha = 2.0$, (**d**) $\alpha = 3.0$, and (**e**) $\alpha = 4.0$.

It can be observed that if $\alpha \leq 1.0$, the components $\Delta_S$ and $\Delta_W$ tend to be similar, particularly at small drifts (before the yielding point of the system); therefore, neither the shear nor the wall distortion components predominate in the total drift, suggesting that a combined mode is achieved. On the other hand, for $\alpha > 2.0$, the response is different no matter the value of this factor. The total drift is mainly related to the wall distortion deformation, while the sliding component remains elastic. At the ultimate state of the system, the $\Delta_S$ contribution to the roof's lateral deformation is around 20%. This result suggests that to reach a wall-distortion-controlled mode (racking-dominant behavior), the overstrength ratio between the in-plane wall load capacity and sliding resistance needs to be at least equal to two. The mode 1 behavior is not attained in the studied cases. Thus, this hypothetical pure sliding mode appears to be improbable unless the sliding capacity is much less than the in-plane wall capacity, particularly when frictional effects are taken into account.

Moreover, it is important to notice the significant difference between mode 2 and mode 3's lateral drift responses. The deformation capacity of mode 3 (e.g., $\alpha = 2$) is larger than that obtained under the mode 2 response (e.g., $\alpha = 1$), suggesting that if a ductile behavior is required, the design must pay special attention to the distribution of the lateral capacity, and a wall-distortion-controlled response must be intended. This result suggests that the overstrength requirements of Eurocode 8 [46] for the design of connections between horizontal and vertical elements is essential for the achievement of suitable seismic performance. However, the ductile behavior is not substantially improved if $\alpha > 2$.

In addition, the pushover analysis results of the M3-1 and M3-2 models are employed to examine the effect of the lateral load capacity distribution on the height in the story displacements. In this case, $\alpha = 2$ is considered for all stories in order to promote a wall distortion mode (mode 3). Various pushover load patterns are employed for both principal directions of the structure, although only the results using a lateral load distribution proportional to the first mode's shape acting on the transverse axis are reported in this paper, because all cases showed similar results.

As presented in Figure 8, when the story load capacity is uniform in all stories (M3-1, $\beta =$ uniform, Figure 8a), the drift demand tends to be nonuniform and the global lateral deformation is concentrated at the first story. At the ultimate state, the model reaches a roof drift of 1.5% but the inter-story drift on the first floor is 2.5%, while on the third floor it is just 0.6%. This situation suggests that if the load capacity is not reduced according to the force demand of each floor, a soft-story failure mode and irregular damage distribution are promoted. In contrast, if the story capacity decreases according to the seismic demand (M3-2, $\beta =$ variable, Figure 8b), it is possible to balance the lateral displacement of the floors and the damage, providing the ductile behavior of the system. Thus, for the same level of roof drift of 1.5% at the ultimate state, the contribution of each floor to the lateral deformation is balanced and no damage concentration is obtained.

Furthermore, the results in Figure 8 are consistent with the findings of Perry et al. [47]. They show that the triggered failure mode is a soft-story mechanism when the story load capacity is uniform in all stories, which is the same behavior that can be observed in Figure 8a for the $\beta =$ uniform case. In addition, the cases studied by Perry et al. [47] for the variable load capacity in height show the same trend presented in Figure 8b for the $\beta =$ variable condition. When the story load capacity is variable, the lateral deformation is balanced between stories, but the largest displacement occurs on the upper floors.

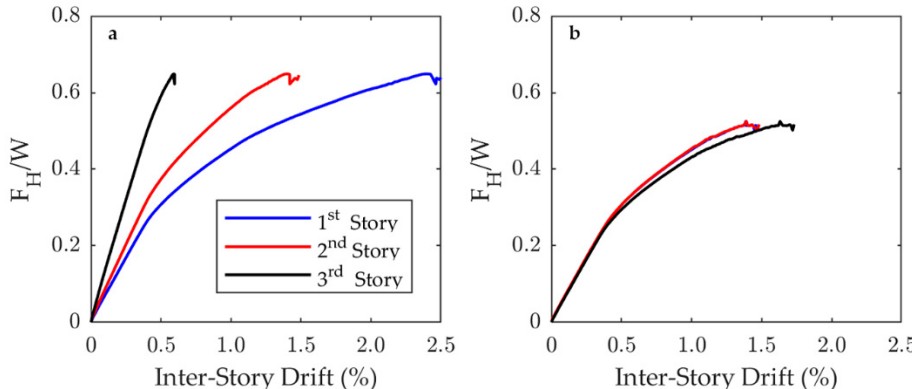

**Figure 8.** Pushover analysis results for the M3 model: (**a**) inter-story drift distribution with a constant lateral load capacity in height ($\beta$ = uniform) and (**b**) with a load capacity proportional to the story force demand ($\beta$ = variable).

### 3.2. Load Capacity Distribution Effect on the Global Performance

The capacity curves obtained for the M5 model analyses are presented in Figure 9, while the global response parameters and the studied cases definition being detailed in Table 4. The yield point is defined as the intersection of the elastic stiffness ($K_e$) with a tangent line to the post-yield zone of the pushover curve. Moreover, the ultimate determined roof drift is defined as the lateral roof displacement when the load capacity drops by 20% or when the nonlinear model fails to find a solution.

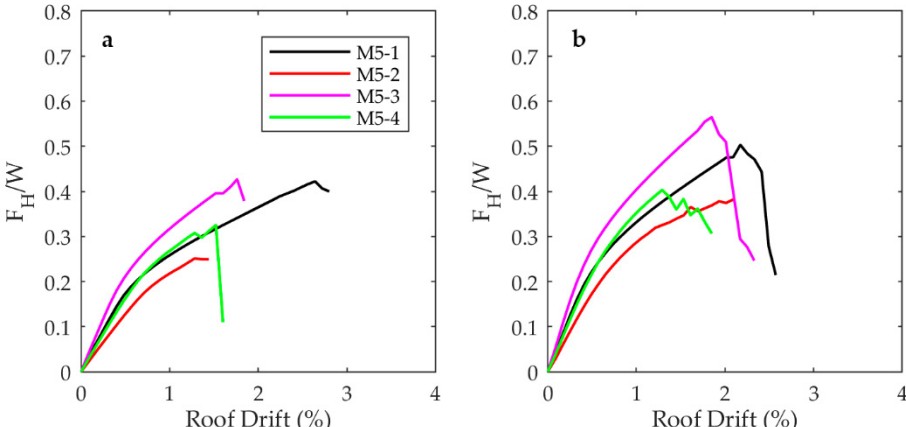

**Figure 9.** Pushover analysis results for the M5 model: capacity curves for the longitudinal (**a**) and transverse (**b**) directions.

As observed in Table 5, a strong relationship between the triggered failure mode and the ductility capacity can be observed. When the wall distortion controls the promoted failure mode ($\alpha$ = 2.0) and a balanced inter-story drift distribution occurs ($\beta$ = variable), the larger global displacement ductility values are reached (4.11 and 4.37 for the longitudinal and transverse directions of model M5-1). As expected, the opposite situation occurs for a combined failure mode ($\alpha$ = 0.5) with concentrated inter-story drift demand ($\beta$ = uniform) (model M5-4 in the longitudinal direction). Furthermore, it can be observed that the triggered failure mode also influences the yield point. When $\alpha$ = 2.0, the structure tends to have a higher elastic stiffness that promotes a yield roof drift between 0.49% and 0.64%. In contrast, for $\alpha$ = 0.5, the yield roof drift increases to the range of 0.60% to 0.81%. Therefore, the results suggest that for combined failure modes (mode 2, $\alpha$ = 0.5), the yielding of the system is delayed but the collapse occurs sooner.

**Table 5.** Summary of the response parameters for the different cases of the M5 model.

| Case | Elastic Stiffness $K_e$ (kN/mm) | Ultimate Load Capacity Ratio $\frac{F_{max}}{W}$ | Yield Roof Drift $\delta_y$ | Ultimate Roof Drift $\delta_u$ | Max. Inter-Story Drift | Global Disp. Ductility $\mu$ |
|---|---|---|---|---|---|---|
| Analysis direction: Longitudinal | | | | | | |
| M5-1 | 7105.2 | 0.42 | 0.64% | 2.62% | 2.71% | 4.11 |
| M5-2 | 4936.1 | 0.25 | 0.60% | 1.43% | 1.65% | 2.40 |
| M5-3 | 9098.9 | 0.42 | 0.50% | 1.83% | 3.05% | 3.68 |
| M5-4 | 6398.8 | 0.35 | 0.71% | 1.59% | 3.57% | 2.22 |
| Analysis direction: Transverse | | | | | | |
| M5-1 | 9332.2 | 0.50 | 0.57% | 2.47% | 2.58% | 4.37 |
| M5-2 | 6725.5 | 0.38 | 0.81% | 2.05% | 2.79% | 2.54 |
| M5-3 | 11,911.2 | 0.56 | 0.49% | 2.07% | 3.07% | 4.26 |
| M5-4 | 8682.0 | 0.40 | 0.61% | 1.84% | 4.33% | 3.01 |

Other relevant aspects of the results shown in Table 5 are the maximum inter-story drift and the global displacement ductility. Although the maximum inter-story drift values for buildings with $\alpha = 0.5$ are on average 8% higher than those for buildings with $\alpha = 2.0$, the global displacement ductility values are quite different. Buildings with $\alpha = 0.5$ have, on average, 38% lower global displacement ductility values than buildings with $\alpha = 2.0$. Again, this confirms the importance of designing buildings promoting a wall-distortion-controlled failure mode.

*3.3. Global Limit States Analysis*

Based on the lateral load-carrying capacity analysis performed on the five-story building, a proposal for damage-limit states is performed. Several authors suggest that important parameters to assess the global damage due to earthquakes are the change in the structural period and the system stiffness. In particular, DiPasquale et al. [48] developed the final softening index ($\delta f$) as an indicator for the global damage state, considering the variation in the first mode period or the stiffness degradation. Complementarily, Ghobarah [49] proposed a relation that defines final softening index limit values for different global damage levels. Using the Ghobarah criteria, four performance limit states are defined according to the roof drift associated with each damage level, stiffness degradation, and final softening index. The considered performance levels are immediate occupancy (IO), operational (O), life-safe (LS), and near-collapse (NC). The use of stiffness-degradation-based limit states instead of common force-degradation criteria is related to the idea of implicitly considering aspects of serviceability and resilience in the established damage states.

Figure 10 presents the roof drift variation for each limit state with respect to the triggered failure mode in accordance with the $\alpha$ factor. A close relationship between the IO performance level drifts and the yield point can be observed for both $\alpha$ values, suggesting that this limit state is attained when the structure begins the incursion in the nonlinear range. Following the same findings of the global displacement ductility (Table 5), it can be observed that for a combined failure mode (mode 2, $\alpha = 0.5$), the low damage limit states (IO and O) are reached at larger global drift values than the wall-distortion-controlled mode (mode 3, $\alpha = 2$), but the large damage states (LS and NC) are achieved at lesser roof displacements. This fact suggests that severe damage and system degradation could be accelerated if the triggered failure mode is not well controlled.

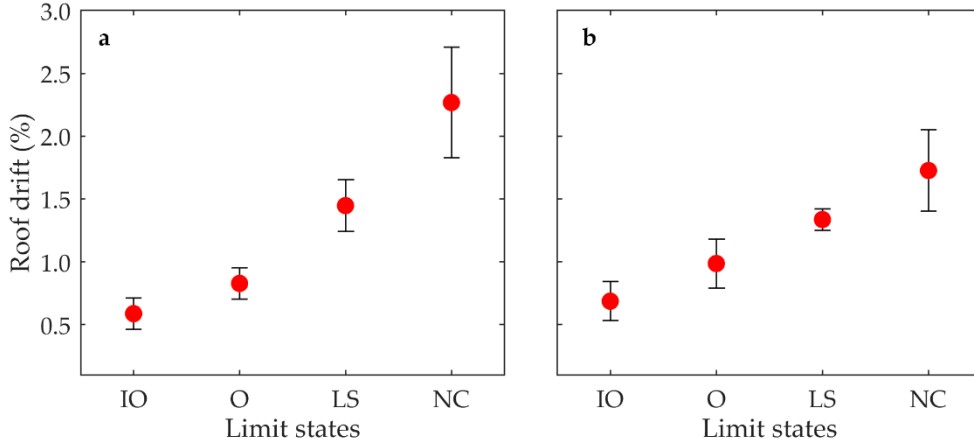

**Figure 10.** Roof drift and limit state relationship according to the triggered global failure mode (red dots are average drifts and back lines show max and min values): (**a**) $\alpha = 2.0$; (**b**) $\alpha = 0.5$.

The average roof drift values obtained for each $\alpha$ factor for every limit state are shown in Table 6. A significant difference can be noticed for the drift limits with respect to other commonly used performance criteria (e.g., VISION 2000 committee [50]), particularly for the low damage levels (IO and O). The results show that in terms of the drift limits for the different performance levels, regardless of the mobilized global failure mode, light-frame timber structures seem to be more flexible and brittle than concrete or steel structures (VISION 2000 limits). In light-frame buildings, the IO and O states are reached later, but LS and NC occur sooner than for concrete or steel structures.

**Table 6.** Average roof drift values for each limit state in accordance with the triggered global failure mode.

| Limit State | Roof Drift Limit According to VISION 2000 [50] | Final Softening Index Limit [49] | $\alpha = 2$ | $\alpha = 0.5$ |
|---|---|---|---|---|
| Immediate Occupancy (IO) | 0.2% | 0.1 | 0.58% | 0.68% |
| Operational (O) | 0.5% | 0.2 | 0.83% | 0.98% |
| Life-Safe (LS) | 1.5% | 0.4 | 1.45% | 1.34% |
| Near-Collapse (NC) | 2.5% | 0.6 | 2.27% | 1.73% |

Concerning the limit states' definitions in terms of inter-story drifts, the results are summarized in Table 7. Comparing the average inter-story drifts achieved for each global damage level with the limit states established in ASCE/SEI 41 [51] and FEMA 273/274 [52] codes, it can be observed that light-frame timber buildings are able to reach larger displacements than code provision for low damage levels (IO), similarly to the results in Table 6. However, for large global damage states (NC), the code-defined inter-story displacements closely match the M5 model results. As can be seen, regardless of the triggered failure mode ($\alpha = 0.5$ or $\alpha = 2.0$), the inter-story drift is near to 3.0%. This situation hints that the lesser roof drifts achieved with $\alpha = 0.5$ are associated with a soft-story behavior. On the other hand, for $\alpha = 2.0$, the system can promote a more balanced displacement distribution.

**Table 7.** Average inter-story drift values for the code-defined limit states with respect to the triggered global failure mode.

| Limit State | ASCE 41 and FEMA 273/274 [51,52] | $\alpha = 2$ | $\alpha = 0.5$ |
|---|---|---|---|
| Immediate Occupancy (IO) | 0.5% | 0.87% | 1.01% |
| Near Collapse (NC) | 3.0% | 2.97% | 3.08% |

*3.4. Fragility Analysis*

The fragility analysis is performed following the results obtained in the previous sections to define the cases to study. For the low deformation capacity model (M3-B), an overstrength factor $\alpha = 0.5$ is used to promote a combined failure mode (mode 2, Figure 4). Moreover, for the ductile model (M3-D), the $\alpha$ factor considered is 2.0, aiming to trigger a wall distortion controlled mode (mode 3). For both cases, a balanced distribution of the inter-story drift among the floors ($\beta$ variable) is the aim by assigning a story load capacity according to the design code's seismic force demand [28].

For the IDA curve constructions, a set of 240 nonlinear time history analyses are performed considering the M3-D and M3-B models in both principal directions, the six seismic records, and the scale factors used. The mean duration of each analysis is around 120 h. The indicator of the structural damage considered is the maximum inter-story drift demand, while the first mode's elastic pseudo-acceleration spectral coordinate is used as the intensity indicator. As the models are segmented into different parallel subdomains, the first mode period is calculated with free vibration tests under initial conditions that excite the first mode but are small enough to keep the response far behind the yielding of the system. The periods obtained are 0.66 s and 0.59 s for the longitudinal and transverse directions of the brittle model and 0.52 s and 0.42 s for the ductile model, respectively. These vibration periods are consistent with the approximated values obtained using simplified models [53]. Moreover, the global displacement ductility capacity values obtained through pushover analyses are 3.3 and 2.4 for the ductile and brittle models, respectively. Figures 11 and 12 present the free vibration response and the IDA curves for the ductile and brittle cases, respectively.

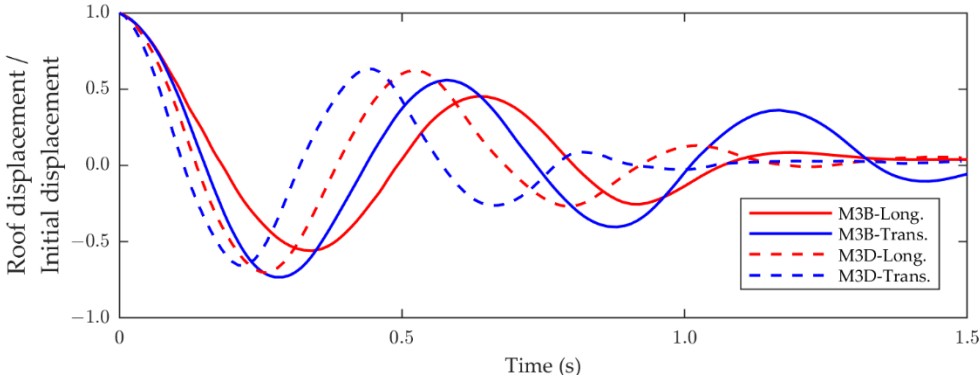

**Figure 11.** Free vibration tests for M3B and M3D models.

Using the IDA results, the fragility curves are calculated for each direction of the M3 structure considering three levels of structural damage (Figure 13). The damage levels are arbitrarily defined in terms of the inter-story drifts ($\delta_{i-s}$), considering 0.8%, 1.5%, and 2.5%. Then, for each of these deformation limits, a log-normal cumulative distribution function is adjusted using the respective pseudo-acceleration coordinates of the IDA curves. The mean and standard deviation values of each adjusted fragility curve are shown in Table 8.

As shown in Figure 13, the failure mode control through the management of sliding and wall distortion mechanisms tends to increase the acceleration demand needed to achieve a certain probability of reaching a defined damage level. This effect is particularly notorious for the large inter-story deformation demands. For the case of the 2.5% of inter-story drift, given the same cumulative probability, the pseudo-accelerations of the ductile case are approximately 30% larger than those of the brittle case. On the other hand, the differences between the brittle and ductile models are not as evident for smaller deformation demands. This situation suggests that by providing a wall distortion failure mode (racking-controlled mode), the level of safety can be improved by increasing the deformation capacity.

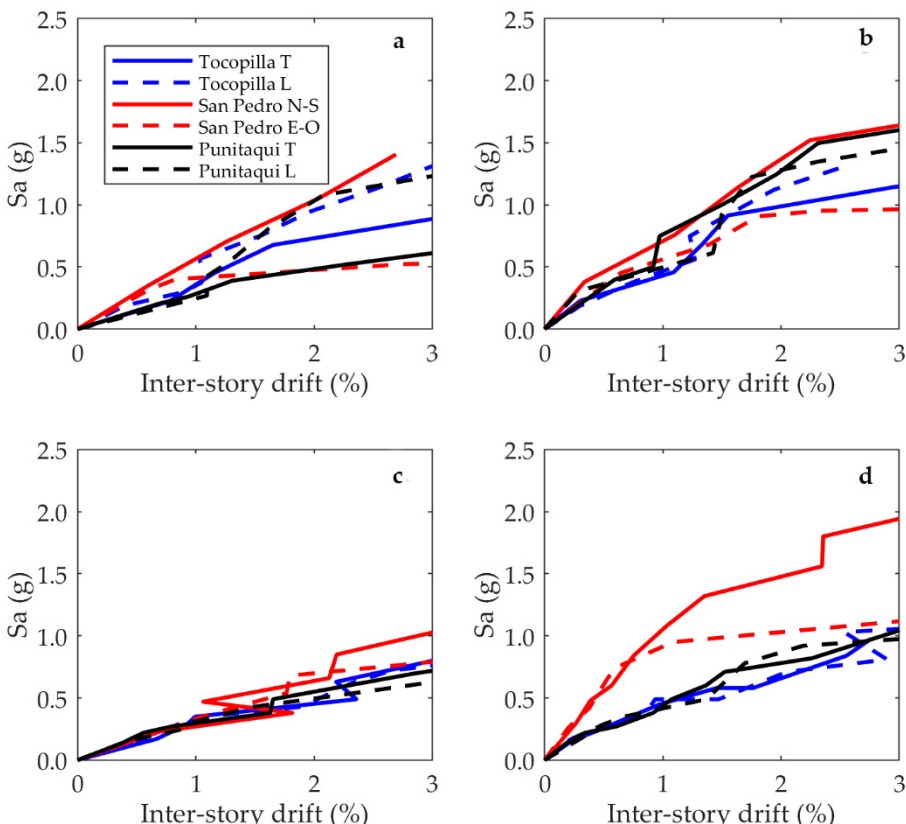

**Figure 12.** IDA curves for the ductile model M3-D in longitudinal (**a**) and transverse (**b**) directions. IDA curves for the brittle model M3-B in longitudinal (**c**) and transverse (**d**) directions.

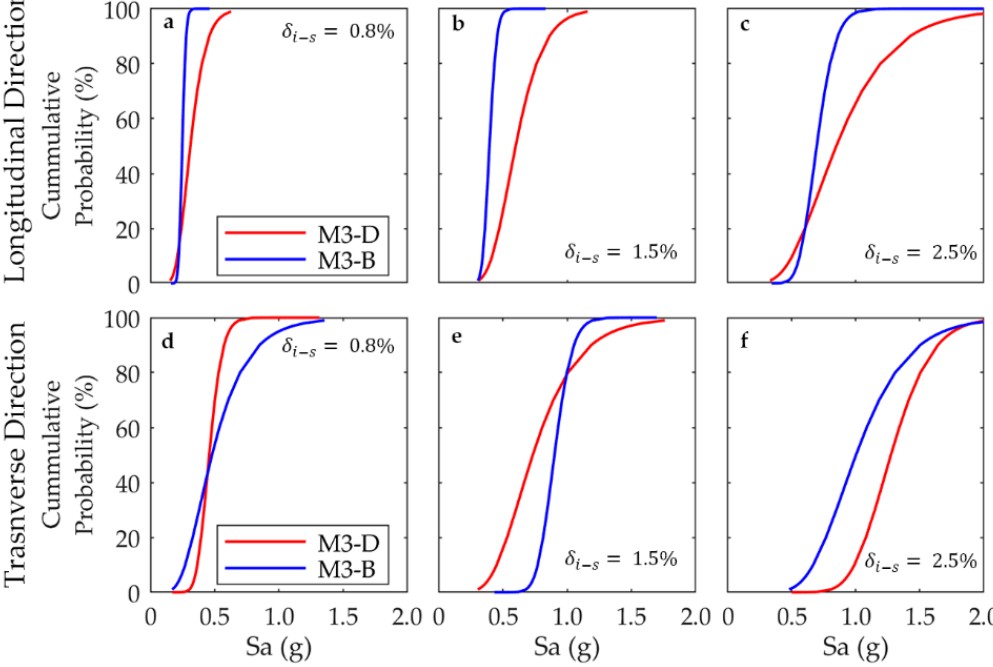

**Figure 13.** Fragility curves for different levels of inter-story drift ($\delta_{i-s}$) for the ductile (M3-D) and brittle (M3-B) cases. Longitudinal direction: (**a**) $\delta_{i-s} = 0.8\%$, (**b**) $\delta_{i-s} = 1.5\%$, and (**c**) $\delta_{i-s} = 2.5\%$. Transverse direction: (**d**) $\delta_{i-s} = 0.8\%$, (**e**) $\delta_{i-s} = 1.5\%$, and (**f**) $\delta_{i-s} = 2.5\%$.

**Table 8.** Mean and standard deviation (between brackets) values of the log-normal fragility curves.

| Case | Damage Level | | |
|---|---|---|---|
| | $\delta_{i-s}$ = 0.8% | $\delta_{i-s}$ = 1.5% | $\delta_{i-s}$ = 2.5% |
| Analysis direction: Longitudinal | | | |
| M3-D | 0.32 g (0.10 g) | 0.62 g (0.16 g) | 0.90 g (0.33 g) |
| M3-B | 0.25 g (0.02 g) | 0.40 g (0.04 g) | 0.70 g (0.12 g) |
| Analysis direction: Transverse | | | |
| M3-D | 0.46 g (0.08 g) | 0.90 g (0.10 g) | 1.29 g (0.24 g) |
| M3-B | 0.52 g (0.25 g) | 0.78 g (0.32 g) | 1.05 g (0.39 g) |

The obtained IDA curves show some differences from the results presented by Pang et al. [14]. They reached pseudo-acceleration levels that are 2 to 2.5 times lower for the same inter-story drifts, suggesting a more brittle response. These differences may be explained by the fact that the predominant failure mode studied in [14] is an almost pure soft story, while in this research the triggered failure mode tends to be more complex. On the other hand, the mean acceleration demands required to achieve the inter-story drift of 2.5% are larger than the mean collapse capacity determined by Estrella et al. [54]. The higher safety levels of the M3-D and M3-B models with respect to the findings of Estrella et al. may be due to the fact that they performed the design using smaller seismic demands and simplified 2D models that disregard the additional capacity that the three-dimensional coupling and vertical load effects can provide.

## 4. Conclusions

This paper has highlighted the importance of using detailed modeling strategies to analyze the seismic behavior of light-frame timber buildings. The results obtained in the modeling of one-, three-, and five-story structures have shown that this modeling strategy effectively distinguishes the most relevant variables in the buildings' seismic performance and can also be efficient if parallel computing techniques are used.

In addition, the findings of this paper show that shear bracket connections and sheathing-to-framing connections control the buildings' responses, as well as the failure mode. This fact suggests that appropriate detailing and distribution of the load-carrying capacity and stiffness between the lateral-load-resisting system components are essential to promote adequate seismic behavior. The structural system has to be provided with a high stiffness and load capacity in the shear connectors to develop a ductile wall-distortion-dominant response. Even though this situation is apparently relevant for the global responses of the structures, the design codes employed in this research do not include specific regulations. Consequently, it seems necessary to advance to a capacity-based seismic design for timber structures to avoid undesired behaviors under seismic excitations.

Regarding the seismic performance of the studied buildings, thanks to the high level of detail of the developed models, a number of specific findings can be stated:

1. The results suggest that an overstrength factor equal to at least 2 ($\alpha = 2$) needs to be considered between the shear brackets and the in-plane wall capacity supplied by the sheathing-to-framing connections if a wall distortion (racking)-dominated response is desired;

2. The fragility analysis results suggest that promoting a racking failure mode can provide higher levels of safety against collapse under large seismic demands. The probability of reaching an inter-story drift demand of 1.5% or 2.5% with a given level of pseudo-acceleration is higher in the model that triggers the wall distortion mode than in the model with a combined failure mode. However, at a lower damage state with an 0.8% inter-story drift demand, the significance of the failure mode control is not clear. Notwithstanding this important result, considering that our fragility analysis

may be limited, future studies should widen the quantity and type of earthquake recordings that are employed;

3.  In terms of the global damage states and system stiffness degradation, the results indicate that the failure mode control may produce a higher displacement capacity for a large initial stiffness reduction. Nevertheless, if the wall distortion mechanism is promoted, the yielding of the system will happen earlier;

4.  Due to the perpendicular wall coupling, the rocking behavior of the walls appears to be less relevant in the global response than the shear sliding and wall distortion; however, further research is required. This particular effect could significantly impact the current design procedures but cannot be properly evaluated using bi-dimensional or simplified models.

Moreover, concerning the development and analysis of the detailed nonlinear models, some relevant outcomes can be summarized:

1.  Seismic performance analyses of a multi-story light-frame timber building were developed through highly detailed models implemented using parallel computing techniques. Using standard desktop PCs with eight logic processors (maximum processor velocity of 4 GHz with 8 GB of RAM), the speed increases achieved for the nonlinear time history analyses were around 2 to 3. This result implies that the time spent running a dynamic analysis was up to one-third of the time required to run the model in a single processor using a sequential scheme. However, an important issue for the computing efficiency was the need to share hard-disk drive space as virtual RAM due to the high level of memory required during the process. Hence, if high-performance computer facilities can be employed using hundreds of processing cores, the computation velocity improvements could be very significant;

2.  Another aspect that can improve the processing efficiency and the reliability of the nonlinear models developed here is the implementation of adaptive and dynamic parallel-domain segmentation techniques, as well as progressive collapse simulation strategies. In this work, collapse management was not performed, and only a static domain decomposition approach was employed because the implementation of more robust parallelization and simulations procedures was beyond the scope of this research. However, in future studies, these two aspect are expected to be developed.

**Author Contributions:** Conceptualization, F.B. and A.J.-C.; methodology, F.B. and A.J.-C.; software, J.C.G. and N.A.; validation, F.B. and A.O.-V.; formal analysis, F.B., A.J.-C. and A.O.-V.; investigation, F.B., J.C.G. and N.A.; resources, F.B. and A.O.-V.; data curation, A.O.-V., J.C.G. and N.A.; writing—original draft preparation, F.B., A.J.-C. and A.O.-V.; writing—review and editing, J.C.G., N.A., A.J.-C. and A.O.-V.; visualization, F.B. and A.J.-C.; supervision, F.B. and A.O.-V.; project administration, F.B. and A.O.-V. All authors have read and agreed to the published version of the manuscript.

**Funding:** This research was funded by CORFO (Production Development Corporation of the Chilean Ministry of Economy), grant number 16BPE-62260.

**Institutional Review Board Statement:** Not applicable.

**Informed Consent Statement:** Not applicable.

**Data Availability Statement:** The data presented in this study are available under request from the corresponding author. The data are not publicly available due to privacy restrictions.

**Acknowledgments:** The dedicated work of undergraduate students Mauricio Salgado, Ignacio Avila, Javier Alarcón, and Luis Pinto is deeply acknowledged. Additionally, the support provided by the Research Group Support Fund provided by the University of Bío-Bío through the INES-92 Grant is appreciated. We also thank ANID BASAL FB210015 for their ongoing scientific collaboration with our academic department.

**Conflicts of Interest:** The authors declare no conflict of interest.

## Appendix A

The structural behavior of wood light-frame systems is strongly influenced by the sheathing-to-framing connections [7]; thus, special care must be paid in the characterization of this joint to assure the reliability of the complex building model response. Therefore, an experimental evaluation of the cyclic and monotonic behavior of these connections was performed to calibrate the constitutive law that was employed in the finite element model.

The sheathing-to-framing connection samples were composed of MGP10-graded radiata pine lumber with a cross-section measuring 35 mm× 138 mm, with a 2.94-mm-diameter and 70-mm-long helical nails and 11.1-mm-thick OSB boards. Each test specimen had four nails (two per side) and was conditioned at a temperature of 20 °C and 65% relative humidity. The tests were performed at the Construction Technologies Research Center of the University of Bío-Bío (CITEC-UBB).

The test specimens were subjected to cyclic and monotonic tests under parallel- and perpendicular-to-grain lateral shear forces, as shown in Figure A1. One monotonic and three cyclic tests were developed for each connection configuration. The cyclic tests were performed under the CUREE Protocol [55]. The displacement imposed on the test specimens ($\Delta$) is defined as a function of the reference displacement ($\Delta_R$), determined as 60% of the ultimate displacement achieved when the load capacity drops by 20% in the monotonic test.

Figure A2a presents the monotonic and cyclic test results per fastener for the sheathing-to-framing joints for both parallel and perpendicular loads. It can be observed that the mechanical behavior under parallel and perpendicular loads is similar for both configurations, but the perpendicular-to-grain response tends to be less ductile. Moreover, before reaching the maximum load capacity of the connection, the two tested setups behave equivalently. However, the load degradation of the perpendicular-to-grain setup is quicker at larger displacements, and the displacement capacity is lower. Additionally, Table A1 shows the average test results for both configurations in terms of the elastic stiffness ($K_{el}$), load capacity ($F_{max}$), displacement at $F_{max}$ ($\Delta_{max}$), yield load ($F_y$), yield displacement ($\Delta_y$), ultimate load ($F_u$), ultimate displacement ($\Delta_u$), and displacement ductility ($\mu$). The results show that the mechanical behaviors of the connections show less than 10% difference between parallel and perpendicular directions, which is expected for small-diameter dowel-type connections according to several standards (e.g., [29,56]). The test results can be found in [34].

Using the experimental data, the pinching4 constitutive law was calibrated in OpenSees, implementing a simple model with a nonlinear spring and two nodes. Due to the similitude between the parallel- and perpendicular-to-grain experimental responses, equal behavior was considered for the model; thus, the parallel-to-grain response was assumed for the modeling of the two directions, and only this component was calibrated. The numerical calibration followed the Benedetti et al. [57] strategy. A comparison between the experimental and calibrated hysteretic curves is shown in Figure A2b, where a good agreement can be observed.

**Table A1.** Average response parameters of tested sheathing-to-framing connections.

| Test Configuration | $K_e$ (kN/mm) | $F_{max}$ (kN) | $\Delta_{max}$ (mm) | $F_y$ (kN) | $\Delta_y$ (mm) | $F_u$ (kN) | $\Delta_u$ (mm) | $\mu$ |
|---|---|---|---|---|---|---|---|---|
| Parallel to grain | 0.95 | 1.88 | 8.27 | 0.94 | 0.99 | 1.50 | 12.09 | 12.21 |
| Perpendicular to grain | 0.80 | 2.23 | 8.35 | 1.01 | 1.26 | 1.77 | 11.34 | 9.04 |

Additionally, to evaluate the accuracy of the proposed modeling approach and the calibrated parameters of the sheathing-to-framing connection hysteretic law, a model of the walls tested in [21] was developed. Figure A3 presents a comparison between the test results and the capacity curve obtained through the model. For both cases compared here (1.2 m and 2.4 m length walls), it can be observed that our model is able to approximate the

experimental response for the entire inter-story drift range (until 7% for the 1.2 m length wall and 6% for the 2.4 m length wall).

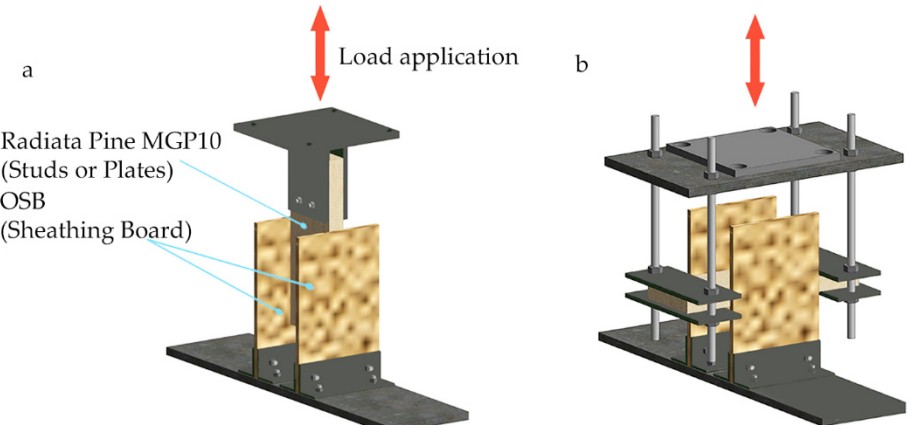

**Figure A1.** Connection tests configuration: parallel (**a**) and perpendicular (**b**) to the grain.

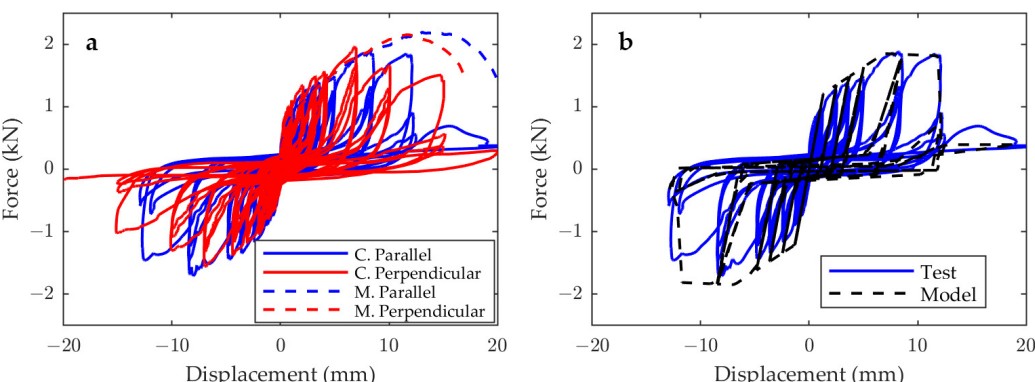

**Figure A2.** Hysteretic behavior of sheathing-to-framing connections: (**a**) parallel- and perpendicular-to-grain mechanical tests; (**b**) numerical calibration using the pinching4 material law.

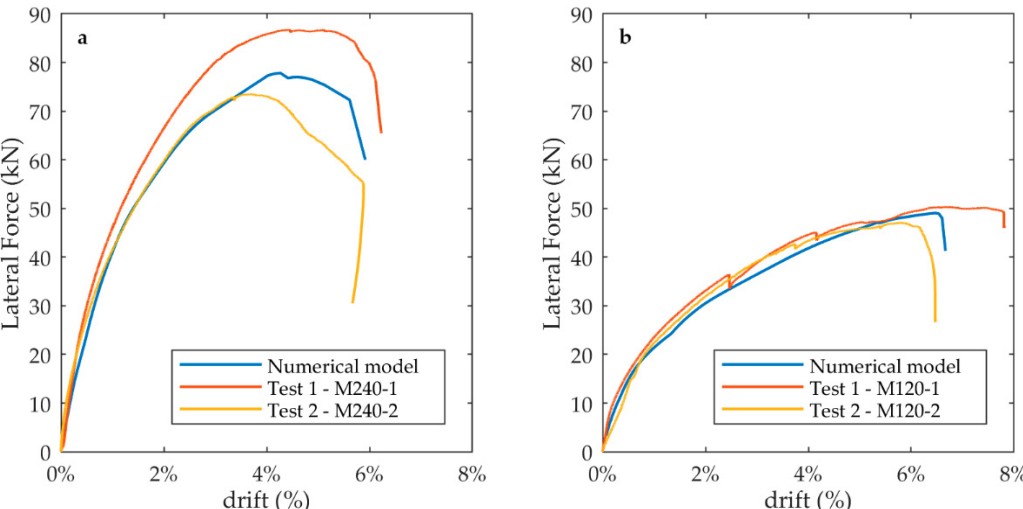

**Figure A3.** Capacity curve comparison for the tested walls reported by Estrella et al., 2020 [21] and the models developed under our approach calibrated using the sheathing-to-framing test results: (**a**) 2.4 m length wall; (**b**) 1.2 m length wall.

**Appendix B**

The parameters of the pinching4 model employed for the connections are presented in Table A2.

**Table A2.** The pinching4 model's calibrated parameters.

| Parameter | Connection Type | | |
|---|---|---|---|
| | Hold Down | Shear Brackets | Sheathing to Framing |
| ePf1 (kN) | 50 | 20 | 1.209 |
| ePf2 (kN) | 99 | 21 | 1.850 |
| ePf3 (kN) | 106.28 | 17 | 1.813 |
| ePf4 (kN) | 35 | 9 | 0.392 |
| ePd1 (mm) | 5.4 | 10 | 1.3 |
| ePd2 (mm) | 13 | 20 | 6.9 |
| ePd3 (mm) | 18.74 | 22 | 11.9 |
| ePd4 (mm) | 24.8 | 24 | 12.2 |
| eNf1 (kN) | −50 | −20 | −1.209 |
| eNf2 (kN) | −99 | −21 | −1.850 |
| eNf3 (kN) | −106.28 | −17 | −1.813 |
| eNf4 (kN) | −35 | −9 | −0.392 |
| eNd1 (mm) | −5.4 | −10 | −1.3 |
| eNd2 (mm) | −13 | −20 | −6.9 |
| eNd3 (mm) | −18.74 | −22 | −11.9 |
| eNd4 (mm) | −24.8 | −24 | −12.2 |
| rDispP | 0.4 | 0.42 | 0.651 |
| fForceP | 0.1 | 0.1 | 0.126 |
| uForceP | 0.08 | 0.01 | 0.042 |
| rDispN | 0.4 | 0.42 | 0.651 |
| fForceN | 0.1 | 0.1 | 0.126 |
| uForceN | 0.08 | 0.01 | 0.042 |
| gK1 | 1 | 1 | 0 |
| gK2 | 0.3 | 0.2 | 0 |
| gK3 | 0.4 | 0.3 | 0 |
| gK4 | 0.3 | 0.2 | 0 |
| gKLim | 0.02 | 0.1 | 0 |
| gD1 | 1 | 0.5 | 0 |
| gD2 | 1 | 0.5 | 0 |
| gD3 | 4 | 2 | 0 |
| gD4 | 4 | 2 | 0 |
| gDLim | 0.02 | 0.005 | 0 |
| gF1 | 1 | 1 | 0 |
| gF2 | 0 | 0 | 0 |
| gF3 | 1 | 1 | 0 |
| gF4 | 1 | 1 | 0 |
| gFLim | 0.04 | 0.001 | 0 |
| gE | 10 | 10 | 1 |
| Damage type | energy | energy | energy |

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
