# Peer review of "Numerical Analysis of the Seismic Performance of Light-Frame Timber Buildings Using a Detailed Model"

_buildings, doi:10.3390/buildings12070981_

Round 1

Reviewer 1 Report

-- Line 81-91: Considering all the uncertainties associated with performance evaluation of light-frame wood structures regardless of the modeling approach, how is the additional complexity introduced due to this modeling approach justified? How much more "improvement" can this modeling approach bring to the final results considering the time required to model the structure, incorporate any changes, model run time and resource required (for example four PCs) ? As mentioned by the authors as well, the simplified models have been validated through a number of tests and accepted by the academics and professionals.   

A more appropriate comparison could be to compare a three story structure modeled using this detailed approach with the simplified model and then compare the results. 

-- Was the pushover force based or displacement based?

-- What were the various patterns used for the pushover analysis?

-- For Figure 6, there is no post peak behavior?

-- Are 6 ground motion records sufficient to obtain fragility curves?

-- Compare your results with the results from this study or other similar studies. 

INFLUENCE OF VARYING STRENGTH, FROM STORY TO STORY, ON MODELED SEISMIC RESPONSE OF WOOD-FRAME SHEAR WALL STRUCTURES

-- Table 6: Instead of roof limit, why not used inter-story drift limits. See ASCE41 and also this might be a good starting point. 

Pang et al-2010-Simplified Direct Displacement Design of Six-Story Woodframe Building and Pretest Seismic Performance Assessment

Author Response

Dear Reviewer. We appreciate your comments and observations. We improved the paper following your suggestions.

In the attached file you can find a document with the answers to every comment.

Thank you.

Reviewer 2 Report

The authors investigate the seismic response of light-frame timber structures via numerical approaches by implementing non-linear static and non-linear dynamic analyses. In particular, the authors have investigated the global failure modes. A major effort has been made to solve computation problems for detailed FE models. The results are interesting and consistent with the state-of-the-art.

Attached there are some minors’ errors founded:

·         Figures: the name of the plots (e.g. a, b, c…) could be in bold or should be put below each figure

Attached there are some major observations/suggestions:

·         Introduction: The state-of-the-art could be improved. The reviewer suggests reading “ Di Gangi, G., Demartino, C., Quaranta, G., & Monti, G. (2020). Dissipation in sheathing-to-framing connections of light-frame timber shear walls under seismic loads. Engineering Structures, 208, 110246.”. In addition, the work presents a dedicated FE strategy developed for OpenSees (https://opensees.berkeley.edu/wiki/index.php/CFSWSWP  “CFSWSWP material”)

·         Line 35: The shaking table test in which damage has been achieved is described in: “ Casagrande, D., Grossi, P., & Tomasi, R. (2016). Shake table tests on a full-scale timber-frame building with gypsum fibre boards. European Journal of Wood and Wood Products, 74(3), 425-442.”

·         Figure 1: insert the one-way floor orientation. It could be helpful to know which shear walls are loaded.

·         Introduction (line 92): Please explain the parametric investigation carried out in section 2.4.1.

·         Line 113: Explain if some capacity design rules are followed to avoid the soft-storey mechanism in the adopted codes.

·         Table 1: insert the reference of the friction coefficient adopted

·         Line 165: Do the shear brackets work in shear and tension? In figure 2, seams only in shear. If the angle brackets have a coupled behaviour, explain how the failure is considered.

·         Line 190: Please explain why the elastic damping is not included.

·         Section 2.4.1: Section 2.4.1 is hard to read. Please simplify the description or put a flowchart/figure explaining the investigation carried out.

·         Line 239: The reviewer is not properly in agreement with neglecting the rocking mechanism when estimating the lateral deformation of LFT shear walls. The rocking mechanism depends on the amount of the vertical loads, the geometry of the shear wall and the properties of the up-lift resisting system (see “Casagrande, D., Rossi, S., Sartori, T., & Tomasi, R. (2012). Analytical and numerical analysis of timber framed shear walls. In Proceedings of World Conference on Timber Engineering, Auckland, New Zealand (pp. 497-503).”). Thus it will be more influenced in the one-storey case than the multi-storey one and/or shear walls with low vertical loads. Please explain in detail the reason for this assumption.

·         Line 251: How the alfa factor is modified is not clear to the reviewer. Is alfa modified by re-design the shear wall with a different number of connections?

·         Figure 8: It could be beneficial to see the elastoplastic curves.

·         Section 3.4: Please explain the IDA increment steps adopted (e.g. PGA increment of 0.1g)

·         Line 457: Put the mean and standard deviation values of the fragility curves in the figure or a new table

·         Conclusion: It could be beneficial to criticise the seismic design code adopted and propose some modifications.

Author Response

(The authors gave the same response as above.)

Reviewer 3 Report

The article addresses an important and very interesting topic of numerical analysis of the seismic performance of light-frame timber buildings using a detailed model, which is appreciated. The article is very interesting and in opinion of the Reviewer has a big potential for the further research. The study include the numerical research. This paper describes the seismic response of mid-rise light-frame timber buildings through a complex and detailed numerical model implemented using parallel computing tools. In addition, the results of this study are expected to contribute to a better understanding of several nonlinear structural phenomena in light-frame timber buildings, such as the effect of lateral load capacity distribution on the global structure. The Reviewer appreciates the efforts done in this paper, however, the Reviewer has concerns regarding to the introduction, numerical modelling, discussion and conclusions. Generally, in this paper the English language should be improved and checked by the Native Speaker (some words and sentences are not clear). In Reviewer's opinion the current version of the paper should be subjected to major revision.

Other comments:

1.     Introduction:

  • Underline please, what is the new of this research/article?
  • What is the difference between this paper and other papers which were cited in the text (more clear)?
  • Please add a few articles about similar researches from this area but please concentrate on the lightweight structure (geodesic dome, roof etc) made of steel or RC. In my opinion the analysis of this kind of structures is similar to research from this paper, also this description will show wide range of applications of this research. Below you can find some papers about lightweight structures under seismic excitations:
  •  https://doi.org/10.3390/ma14164493
  •  https://doi.org/10.3390/app12042116
  •  https://doi.org/10.1016/j.compositesb.2017.11.051
  •  https://doi.org/10.1016/j.tws.2019.03.039

2.     Numerical modelling

Why the Open Sees was used in the numerical research? You can find better software with full graphical interface and many constitutive models of material.

- Please add the density and boundary conditions of finite elements.

- The seismic excitation is very stochastic phenomenon, thus please explain your approach (line 161 – 165) “Besides, for the sheathing to-framing and shear brackets connections, the model is idealized coupling two orthogonal nonlinear springs to take into account the parallel and perpendicular to grain response of the sheathing-to-frame nails, as well as the shear and tension stiffness for the shear brackets.”. Is it true? How can you compare this approach/assumption to real behaviour of the structure under seismic excitations?

-  Please add the value of damping in numerical analysis.

- Why the structure of M2 and M4 were not analysed (according to line 233)?

- Why the shorter seismic records were not used? What was the reason of it?

- Which constitutive material model was used for numerical analysis?

-   The modal analysis was used in numerical analysis?

3.      Description of results:

·       Please explain what  the “drift” in Figure 7, 8, 9 mean?

4.     Discussion:

·       The discussion of results is really poor. Please add the discussion of the results with compare to other similar researches.

5.     Conclusions:

·       What is the general conclusion from this research?

At the end I hope that my comments will be helpful for the authors.

Author Response

Dear Reviewer. We appreciate your comments and observations. In the attached document we present the answers to every comment.

Thank you.

Regards.

Round 2

Reviewer 1 Report

-- Regarding Comment#1 of the previous review about comparison with the simplified model, I disagree with the response at the end. Since this is purely a numerical study (not a full-scale system-level validation like a shake table test or assembly tests), in order to verify the global response, either compare the results with some tests or since the simplified modeling approach is widely accepted, compare the proposed model with a simplified model. The latter seems to be easier since it's difficult to obtain all the information about a shake table test to then verify the results. 

A model like Folz and Filiatrault should be very easy to set up for a simple three-story structure. Or maybe even a one-story to compare with M1. 

-- Regarding Comment#4 of the previous review, this raises some questions about the model being utilized in this study. 

The issue is not only with M1 but same is seen for pushover curves of M3 and M5. With all the complexities and computing time, a model like this should be able to model the behavior well within the nonlinear range. However, in this case, it fails to do so. 

The post peak behavior for a light-frame wall should go well into 7 to 8% drift. For example, in the light-frame wood example of the FEMA P695 report, 7% inter-story drift is used to identify collapse. In the case of M1 ( since it is a one-story structure, roof drift and inter-story drift would be the same) the roof however seems to fail at only around 2%. Let's say the authors were asked to perform a FEMA P695 study, for the static pushover analysis, they'd need at least 80% post peak load to determine some parameters needed in FEMA P695 analysis. 

Also, if the authors perform similar IDA for their examples, how can their model work in the nonlinear range close to 7 and 8% inter-story drift that was used in the FEMA example and is is typically used in other wood studies?

Author Response

The comments of the Reviewer are deeply acknowledged. In the attached file we send the response to every observation.

Reviewer 3 Report

Thank you for your improving. The Reviewer have last questions:

Comment 4: Introduction. Please add a few articles about similar researches from this area but please concentrate on the lightweight structure (geodesic dome, roof etc) made of steel or RC. In my opinion the analysis of this kind of structures is similar to research from this paper, also this description will show wide range of applications of this research. Below you can find some papers about lightweight structures under seismic excitations:

https://doi.org/10.3390/ma14164493

https://doi.org/10.3390/app12042116

https://doi.org/10.1016/j.compositesb.2017.11.051

Response: We acknowledge the observation. In the Introduction revised version of the paper, we have added a brief analysis concerning other light weight structural systems (lines 107 to 112). However, we focus our analysis on other platform-like structures with an equivalent seismic behavior as light-frame timber buildins, with similar failure modes and deformation mechanisms. Hence, the references recommended by the Reviewer were not included because these structures display other specific mechanisms. For example, in light-frame steel buildings (https://doi.org/10.1016/j.tws.2019.03.039), it is exhibited certain local effects that can control the global response but are distinct from the behavior of light-frame timber structures (e.g., local buckling in frame elements, diagonal straps yielding). Since these particularities, the seismic analysis through nonlinear models may demand other particular strategies that are not required for the study of timber structures.

Additional comments of Reviewer: The Reviewer would like to ask you to underline the aspect of seismic response (obvious seismic response of lightweight structures) and numerical analysis with Time History. Thus, in the proposed papers, these aspects were included.

Comment 7: Numerical modelling. The seismic excitation is very stochastic phenomenon, thus please explain your approach (line 161 – 165) "Besides, for the sheathing to-framing and shear brackets connections, the model is idealized coupling two orthogonal nonlinear springs to take into account the parallel and perpendicular to grain response of the sheathing-to-frame nails, as well as the shear and tension stiffness for the shear brackets.". Is it true? How can you compare this approach/assumption to real behaviour of the structure under seismic excitations?

Response: Given the behavior of timber frame structures under seismic loads, the mechanical response of fasteners and connections is a complex multidimensional phenomenon (e.g., Kuai et al. 2022 https://doi.org/10.1016/j.engstruct.2021.113599). For the sheathing to framing connection, Källsner and Girhammar 2009 (https://doi.org/10.1617/s11527-008-9463-x) showed that the parallel and perpendicular to grain shear are demands that act simoultaneously, particularly in the corner fasteners of the sheathing board. In adition, for the shear brackets, it has been observed that the coupled response between shear and tension occurs because the connector device is located in zones of the structure where the structural elements tend to uplift and slide (Shen et al. 2013 https://doi.org/10.1016/j.conbuildmat.2013.07.050). Besides, these brackets can provide large stiffness and strength in both loading directions (Liu and Lam 2018 https://doi.org/10.1016/j.engstruct.2018.05.013).

In consequence, the model we develop must provide load-displacement response in the principal directions of the actions for the sheathing-to-framing nail connections and the shear brackets devices."

Additional comments of Reviewer: The Reviewer cannot see response of “How can you compare this approach/assumption to real behaviour of the structure under seismic excitations?" Please answer the above point.

Finally, I hope that my comments will be helpful for the authors.

Author Response

(The authors gave the same response as above.)

Round 3

Reviewer 1 Report

The authors have addressed the reviewer's comments and the manuscript is now suitable for publication.